# The Opportunities and Risks of the Soil Security Metaphor: A Review

**Catherine Allan** [1,2]

1   Graham Centre for Agricultural Innovation (Charles Sturt University and NSW Department of Primary Industries). Charles Sturt University, P.O. Box 789, Albury, NSW 2640, Australia; callan@csu.edu.au
2   Cooperative Research Centre for High Performance Soils, Callaghan, NSW 2308, Australia

**Abstract:** Language both represents and constructs our reality. Soil Security is a proposed new frame for encouraging the systematic and interdisciplinary approaches to soil research needed to address complex needs. This bold attempt at reframing may, however, have numerous consequences in addition to its central intent. This review paper explores words and discourses related to 'Soil Security'. Current understanding of how language frames reality is presented, emphasising the roles of metaphor and entailments. Soil Security is then situated in relation to the broader construct of Environmental Security, with references to security of water, food and energy. Against this background, aspects of Soil Security are explored, and some cautions issued to users of the term. The soil science community is urged to actively consider the implications and nuances of any discourse, including that of Soil Security, with which it engages, or risk being led to operate in unintended or unwelcome ways. To guide this engagement, the review paper concludes with suggestions on how to reflect on the practice of soil science and its role in the future of humanity.

**Keywords:** soil security; metaphor; discourse; framing; sustainability

## 1. Introduction

'Soil Security' is a recently proposed frame for encouraging the systematic and interdisciplinary approaches to soil research needed to address complex needs, such as those behind the United Nation's Sustainable Development Goals (SDGs) [1]. This active reframing of practice is a response from some soil scientists who sense that, despite documentation of the rate and impacts of soil degradation around the world, complex soil information is often excluded from global modelling and forecasting [1,2]. It is particularly noted that soil, although closely related to at least five of the SDGs, was not 'recognized with its own SDG' [3]. At a symposium called in 2012 to address the poor recognition of the soil sciences in the global response to human-induced change, conscious use of the term 'Soil Security' was suggested to 'focus the soil effort', and to be more integrating and active than existing formulations such as 'soil protection' [4]. At that symposium, Soil Security was agreed to refer to 'the maintenance or improvement of the world's soil resource so it can provide sufficient food and fiber, fresh water, contribute to energy sustainability and climate stability, maintain biodiversity and overall environmental protection and ecosystem services' [4]. The concept of Soil Security is considered to 'make explicit connections between soil and the other global existential challenges' and to highlight the importance of maintaining the range of soil attributes that support and regulate biophysical systems [5]. While this integrating, sustainable intent remains, the presentation of the concept of Soil Security has been refined such that the core security goal is described as encompassing the five domains of capability, condition, capital, connectivity and codification [6]. Based on the definitions provided in [6], *capability* relates to the functions that a soil can be expected to perform, in the context of its own inherent, or reference state. *Condition* refers to the current state of the soil and is

considered as the change, if any, in capability compared to the reference state. While capability relates to soil characteristics such as profile and form, which change only slowly over time, soil condition is measured on a short-term management time scale. The economic value of soils is captured in the *capital* dimension, with emphasis on natural capital. Invoking capital recognises soil as an asset, or stock, that enables services (to the environment, especially humans in the environment) to be produced. Some social dimensions of soils are captured under *connectivity*, which has a focus on communication, knowledge sharing and governance. *Codification* supports the other dimensions, especially connectivity, through engaging with policy and regulation.

A second Soil Security Conference, held in Paris in December 2016, and involving many of the originators of the Soil Security label, aimed to consolidate the nascent concept by connecting soil scientists with policy makers, practitioners and a wider range of researchers [7]. A number of papers at the Paris conference confirmed that, while the concept of Soil Security has some equivalence with traditional, biophysical-centred soil conservation practice, it is more inclusive of social, economic and policy aspects of soil [8].

Use of the term 'Soil Security' is not widespread, but appears to be increasing. For example, within the Scopus database of journals, the first instance of a paper using 'soil security' is in 1989, but then it related to laboratory-based stability tests of internal erosion of soil samples. Use of the term Soil Security to indicate protecting the soil resource, and prevention of soil loss or degradation, first appears within the database in a single paper from 2011. Since then up to the time of submitting this review, there have been 70 papers published in Scopus-indexed journals that invoke Soil Security in the sense of protecting the soil resource. Nearly half (33) of those were published in 2018 and the first half of 2019. The 3rd Soil Security conference held in Sydney in December 2018, along with this Special Issue, are likely to encourage more use of the term. Now is, therefore, an excellent time to consider the implications of continuing to promote the term and concept among the soil science community.

It is clear what Soil Security means to those initiating use of the term. Past experience with the complexity of language and cognition (see, for example [9]), suggests this bold attempt at reframing may, however, have numerous consequences in addition to its central intent. These potential consequences deserve scrutiny within the disciplines of soil science. The active intent to change the global view of soil science via a two-word term is open acknowledgement that words can influence thought, practice and policy. It would be disingenuous to deliberately use words as change agents, then dismiss any reflection on the potential consequences of the act as side-tracking into semantics. Rather than being a distraction, reflection on the words used by soil scientists is essential for those who seek to successfully position soil as a key consideration for global governance and management.

In this review paper, I explore the words and discourses related to 'Soil Security'. First, because the act of re-framing was deliberate, I outline some of the current understandings of frames, emphasising the roles of metaphor and entailments. I then situate Soil Security in relation to the broader construct of Environmental Security, with references to security of water, food and energy. Against this background, aspects of Soil Security are explored, and some cautions issued to users of the term. I conclude the paper with suggestions on how to reflect on the practice of soil science and its role in the future of humanity and global environments.

## 2. Discourse; Frames, Metaphor and Entailments

Humans have particular strengths in systems of shared cultural tradition, one aspect of which is language [10]. Language both represents and constructs human understanding of our world [11]. An important act of language is to frame the lived experience of individuals and societies, with some temporal and geographical stability. Humans understand and relate to the world around them with the help of these frames, or mental knowledge structures, that mediate what is observed [12]. One way the frames are developed and maintained among humans—socially, culturally and institutionally-is through discourse, the shared use of words and symbols. The human ability to view one idea through another [13] enables the use of metaphor within discourse to help construct and maintain

meaning. Metaphors enable an idea being communicated to be understood through reference to a source idea [10]. Because an idea (usually called the target idea) can be understood through another, or 'source' idea, metaphorical phrases usually contain some form of the verb 'to be', for example, Life is a Journey [14] or Nations are brothers [15]. Metaphors are part of the way humans communicate, but they are not simply a tool for passing on information. Metaphors help to shape how we think, and how we make sense of the information around us. The Cognitive Theory of Metaphor (CTM) posits that metaphor has this particularly important role in the creation and maintenance of meaning from words and images because metaphorical speech reflects metaphorical thinking [16]. CTM also suggests that metaphorical expressions cluster together in a systematic way that reflects underlying conceptual structures [17]. Metaphors structure understanding by enabling conceptual 'mappings' between source and target ideas, achieved through systems of *entailments*, the mental associations between corresponding elements of the concepts in metaphoric relation. Entailments enable people, individually or in groups, to use selected understandings from a source idea to interpret and understand related target concepts [18]. Building on one of the examples above, a response to a query about a person's health that says 'she is travelling a hard road . . . ' makes sense because things to travel by, such as roads, are an entailment of the Life is a Journey metaphor. A road is just one of many potential entailments; when people's lives are 'becalmed', the entailments of sea journeys are invoked, or if someone's 'life is on track', ideas from train journeys, including planning, orderliness, clear direction and scheduling, map onto the target idea. Humans continuously use framings and entailments as they accept and reject explanations of the world around them. This cognitive-centred approach suggests that words do not have meaning in themselves, rather, that meaning is created in the minds of people sharing the words, with reference to the frames and entailments built over time [19]. This is not to suggest that new ideas cannot be conveyed through the use of metaphor in discourse—the act of creating meaning in the mind has many influences alongside cognitive mapping [20]. Metaphor is part of both communication and cognition [21], and to understand how meaning is being created and shared requires a deep engagement with discourse [22]. The remainder of this paper engages deeply with the discourse of Soil Security.

## 3. Engaging with the Soil Security Discourse

Firstly, it is useful to note that the term Soil Security contains two words. The first—soil—is very familiar, being used with other constructions such as soil conservation, soil protection and soil degradation, and its meaning is so obvious that it appears there is nothing more to say about the 'soil' in Soil Security. What is being communicated by using the word soil, however, does warrant consideration, as its use is a deliberate choice from among a number of possibilities. While dirt, mud and soil are all commonly used in general conversation, soil is the word almost always chosen by scientists when labelling themselves and their work; for example, there is an International Union of Soil Sciences, and the various English language universities around the world teach courses in soils, rather than courses in dirt. This choice is underlined by the University of New England which, while, promoting its soils course on-line, notes that, "The thin skin of soil that mantles the land surface of the earth is a complex and fascinating chemical, physical and biological system. Far from being mere dirt beneath your feet, it is teeming with life "(UNE 2019). In contrast, 'dirt' is rarely used in academic writing unless the author is being clever with the title or seeking to get down with the people, for example through popular media such as Dirt! The Movie [23], or the adventures of dirtgirl [24]. Similarly, when an Australian Minister choses to welcome readers to a Landcare Australia website by stating ' In my short time as Minister for Agriculture and Water Resources I have kicked the dirt with as many of our farmers as possible' [25], it is presumably a deliberate choice that emphasises empathy with real, non-theoretical, issues.

The difference in the use of soil and dirt may be because dirt, the more colloquial term, has more everyday negative associations, as reflected in the phrases . . . 'dirt poor', 'treated like dirt', and 'get the dirt on'. Similarly, we all know that when 'mud is flung' that 'mud sticks'. Soil also has some negative

connotations, but they tend to be milder and literal rather than figurative; having a laundry literally full of soiled clothing is less problematic than metaphorically hanging your dirty linen out in public. It is also possible that in choosing 'soil', scientists are following a long academic tradition of favouring Latin and Greek words to describe their work. The etymology of dirt and mud is via Old Norse [26] and Old German [27], respectively, while the word soil comes into the English language later, from Latin via French [27]. English academic writing frequently borrows from Latin—its original printed language—because doing so confers some prestige to the work [28]. Scientists in general, also eschew the vagueness of the colloquial for the apparent precision of academic language [29]. Whatever the ultimate reason, in choosing to use the word soil over alternatives such as dirt, soil scientists effectively situate the discourse about its management and protection in the domain of experts. Use of the term Soil Security, along with soil conservation and soil protection, is a subtle but constant reinforcement of an academic, expert claim on the discourse, including a claim on who should describe both the problems and their solutions.

The other word of the term is security, which has both communicative and metaphorical complexity. Understandings and uses of security have developed since its appearance in writing in English in the early 1500s. Originally referring to the condition of being secure, that is, of being free from care or apprehension [27], secure and security now have multiple definitions and meanings (see, for instance, the entry for 'secure' in any version of the English Oxford Dictionary). In common conversation, security is used in three general areas—protection of property, reducing apprehension for the future and autonomy of action—each examined briefly below.

The first, everyday understanding of security relates to conceptualisations of ownership and protecting what is owned. High Street security firms, for example, undertake to protect premises from damage, intrusion or theft. These actions make sense within a shared framing of property, such that ownership confers rights to exclude others [30]. Understanding of property varies among communities and over time, but it always involves conveying rights constructed and supported by social structures such as laws, regulations, norms and/or traditions. Ideas related to private property and ownership appear to have arisen with agrarian societies, and relate to the new needs of sedentary life. This highlights that property rights are socially constructed tools for individuals or societies to achieve and maintain particular types of order [31]. Security, in this context, is about reduced anxiety of loss or damage achieved through creating defences, surveillance of those defences, and the threat or actuality of punishment if the defences are breached.

The second general use of the term security is closest to the historical meaning, that being secure is to be alright in the future, safe from care and need. Here, there is a sense of probabilistic calculation and risk mitigation in the notion of security, and of exercising control measures to manage calculated risks. This aspect of security is reflected most clearly in everyday discussion of financial security; for example, superannuation documents regularly refer to security as planning to ensure there is sufficient money to cover needs in the future, and the 'peace of mind' that sufficiency brings with it (cue images of a silver-haired couple laughing on a beach). In this conceptualization, security is about being safe from the threat of known and unknown future risks, achieved through calculating and controlling risks and, sometimes, by creating opportunities.

The idea of national or state security builds on aspects of maintaining possession of what is owned, and planning to cover the needs of now and the future. It takes the notion of security beyond individuals and small communities into broader levels of governance by naming and addressing threats to sovereignty. National security requires territory and staked boundaries [32], as well as the power and capacity to act for and within those boundaries. Both internal and external measures are applied for National Security, and include using diplomacy to isolate hazards, making allies and having adequate armed forces and intelligence agencies [33]. Security in this National or state sense is very much about control and defence.

In recent decades, the word security has been applied to securing and safeguarding a particular type of property—'The Environment'. This is a complex mapping of ideas, as Environmental Security

is considered an integration of earlier ideas of environmental protection, or sustainability, and National Security [34]. Environmental Security features prominently in the World Commission on Environment and Development's report from 1987 [35], and for a period, was an important part of the political discourse of the United States [36]. The formulation of Environmental Security was a response to globalisation and the realisation that economic activity was impacting on global systems in discernible and problematic ways. Part of the discourse of Environmental Security is the suggestion that environmental degradation, which imperils the health and safety of citizens, requires nations to provide similar responses to those made toward traditional military threats [37]. The transformation of an issue into a matter of security through words is referred to as securitization [38]. Although Environmental Security appeared as a term in the 1980s, environmental 'securitization' can be traced back to the 1960s, the era of Silent Spring [39] and the images from the Nasa's Apollo mission that showed the boundaries of a fragile Earth, leading to growing public awareness of human impacts on the environment, and an obligation to act for the global biosphere [40].

Once security was linked with the environment, securitization language in relation to sustainability became common. Related formulations include Food Security, which is focused on sufficient and equitable nutrition for the growing human population [41] and Energy Security, which has a similar formulation [42]. Water Security is also about humans needs [43], however, constructions of Water Security include specific references to environmental needs, so that by 2012, Water Security, as a term, was posited as replacing the 'out of fashion' term, sustainability [44]. Water Security fits easily with sovereignty and securitization, as water moves across jurisdictions and can be readily categorised as property using classic property right approaches. Using security in relation to water produces more than water markets and trading, however; 'water security' invokes the idea that water is an existential issue, dealing with threats to the sovereign rights of the state and the freedom of the individual [45]. Water Security, as with Environmental Security, can thus be seen as an issue for high politics and a justification for the use of power, control and force [45].

The link between National Security and Environmental Security is considered a powerful communicative device [46], and some of that power is because Security is a metaphor. The existential threat of human invasion and the potential for environmental degradation are rendered as congruent, and the military ideas map from the former to the latter. Discourse involving the metaphor will invoke entailments of protection, and safety, aspects of ownership, control, power, and force.

## 4. Soil Security as Metaphor

As noted above, promoting the use of Soil Security was a deliberate attempt to include soil science and soil scientists in the global discourse of Environmental Security. It should be no surprise that Soil Security is also a metaphor, or indeed a cluster of metaphors, and the consequence is that the meanings created through discourse cannot be considered simple or static. There is real potential for securitization of soil, following a similar pathway as that described in the case of water security. Soil does not, generally, move across jurisdictions, although erosion can and does place soil and nutrients in water, and air columns that may cross jurisdictional boundaries. In addition, there are complex linkages between soil, environmental scarcity and conflict [47].

The proponents of Soil Security have addressed the potential for multiple, possibly risky, understandings by defining Soil Security through the five dimensions of capability, condition, capital, connectivity and codification [2]. As described in the introduction, the capital dimension emphasizes the economic framing of soil—referring to it as an asset or a stock implies a socially constructed understanding of ownership and property. The capital dimension, therefore, has clear links with the security metaphor of owning, and the need to protect what is owned. Capacity relates to the thing that is owned, the basic functioning of a soil in relation to its reference state, and condition draws attention to contemporary changes in capacity. The focus on change, and on trying to preserve an ideal state suggests these dimensions are closely related to the 'being safe' framing of security. Codification and connectivity draw attention to the need to control social and institutional factors to be able to keep

what is owned (the soil) safe. In short, the related metaphors of Soil Security are that Soil Security is being safe, owning and being in control (Figure 1).

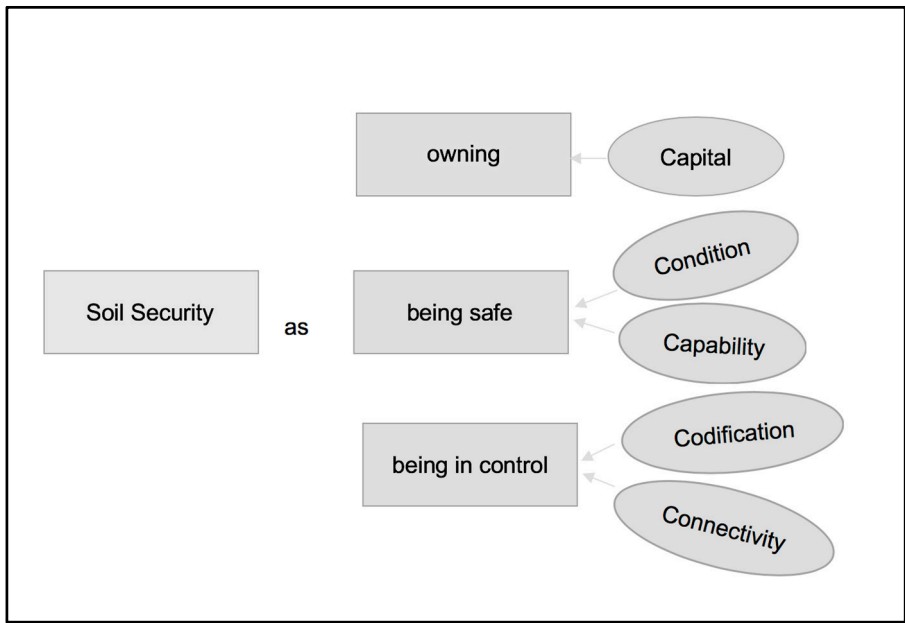

**Figure 1.** Three related metaphors of Soil Security, and their relationship with the dimensions of Soil Security as articulated in [6].

The metaphors of Soil Security create a discourse of keeping soil safe into the future by being in control. This reflects the stated intentions of using security to frame discussions of soil [4], but there are other implications of embracing Soil Security.

## 5. Implications of Actively Embracing the Soil Security Metaphor

When *I* use a word,' Humpty Dumpty said in rather a scornful tone, 'it means just what I choose it to mean–neither more nor less.'

'The question is,' said Alice, 'whether you can make words mean different things–that's all.'

'The question is,' said Humpty Dumpty, 'which is to be master–that's all' [48].

Which, indeed, is to master? By actively situating soil as part of the broader discussion of Environmental Security, parts of the soil science community are drawing attention to the serious consequences of soil degradation for humans and their environments. Using metaphorical language in this way is a rational act, as metaphors have been shown to provide the decision premises for political decision makers [49]. The Security metaphor has also been shown to open up previously narrow approaches to management and policy, facilitating discussions on human values, ethics and considerations of power [50]. However, because of the multiplicity of meanings created through communication, soil scientists need to be aware of, and seek to manage, the full influencing potential of their chosen metaphor. For example, whether welcome or not, securitization is an entailment of the overarching Security metaphor into which soil is now included. Using the Security metaphor to influence policy becomes a double-edged sword (so to speak), as soil issues do indeed become part of the global conversation, but because it is a large conversation, those soil issues may become diluted in broader debates about National Security and sovereignty. This dilution poses the risk that policy makers will not be able to prioritise soil with so many competing security interests [38]. Considering environment protection in terms of security enables new and differing constructions of what the 'environment' is in relation to humans. The environment, in part or whole, can be labelled as a critical

infrastructure, in need of securing against threats of disruption and failure. While securitization adds weight to calls for urgent environment intervention, the lens of securing critical infrastructure against catastrophe risks obscuring everyday, minor disruptions [51]. Pressing the urgency of protecting soil by emphasising its critical importance to humans may, in a securitized framing, divert attention from the gradual degradation which characterises most concerns with soil.

Another important implication of the Security metaphor is that many of its entailments reinforce conservative, protectionist and/or risk-averse approaches to soil science, soil management and soil governance. Avoiding risks may mean avoiding opportunities, including opportunities for novel uses of the soil resource, adaptation, change, and transformation. The question of ownership is also foregrounded by the Soil Security metaphor. Who 'owns' the problem and solution? A partial answer was provided earlier, with the science community claiming some of the ownership by showing preference for the elite word 'soil', and through centuries of description, naming and mapping. Ownership of physical soil is also messy; although globally, much agricultural, forestry and mining land is in private ownership, the Security metaphor brings into focus the large public good component of the soil resource.

It is not inconceivable that a focus on the codification dimension of Soil Security, coupled with protectionist risk aversion, could support and even nurture rigid, inflexible, reductionist approaches to governance. Such reduction and risk aversion are the antithesis of the systemic approaches needed to govern natural resources in the Anthropocene [19]. Indeed, reductionism and risk aversion are part of the issue that the formulation of Soil Security seeks to address.

None of the desirable or undesirable potential futures enabled by the Soil Security metaphor are inevitable, but neither are the likely trajectories predictable. The worst thing that soil scientists can do, now that the Security metaphor has been invoked, is to leave its future to others, and concentrate on the science of soil alone. To engage with the multiple meanings that will be created through use of a powerful metaphor, soil scientists must be prepared to engage in mutually helpful interaction, through transdisciplinary teams [52]. The recognition that the connectivity dimension of Soil Security 'brings in a social dimension around soil' [6] can be understood as an opportunity to broaden the soil sciences into these transdisciplinary spaces. Soil scientists should be questioning boundaries, including with whom to work, who to learn from, and how best to communicate [53]. That this is not easily achievable is demonstrated by the papers presented at the 2nd Soil Security conference in Paris. Despite this gathering being expressly about connection, analysis of the coverage of the five dimensions of soil security revealed that capability and condition were the most addressed, followed by capital, codification and, lastly, connectivity [7]. While expansion of practice of soil science to include actors from policy and social spheres may be intimidating, and unwelcome to some, to not expand into transdisciplinary practice risks losing control of global soil stories. The result may be more problematic for soil science than the original issues of not being part of the global story telling.

To ensure that using the term Soil Security broadens rather than narrows the global discourse, soil scientists need to fully embrace all five dimensions that underpin the term. To do so is to accept not only that it means more than, say, soil conservation, but also that the behaviours of scientists, managers, practitioners must change. Fortunately, there are many examples of interdisciplinary practice, particularly from the water and climate sciences, upon which soil scientists can draw. Various forms of collaborative inquiry and co design (for example [54]) enable systems to be considered when articulating the issues to be addressed and how to approach them. Reflexive approaches to the practice of research and communication also encourage integrating and interdisciplinarity. For example, the shared act of interrogating and understanding a narrative or discourse can build collaborations, share understandings and lead to the design of new futures [55]. Innovative arrangements for governance [56] and for moving beyond single issues [57], become more attainable with these collaborative, transdisciplinary approaches. For transformation of practice, collaboration needs to occur not only among scientists of different disciplines, but also at science and practitioner interfaces [58]. Distilling the lessons from these and related texts, I suggest that soil scientists who consider themselves to work for 'Soil Security' should:

- Continue to focus on the five identified dimensions of Soil Security rather than allowing the headline term to do the communicative and framing work. 'Soil Security' may be a catchy shorthand for what is required, but the entailments of metaphors of security are so complex, contradictory and deeply embedded in human history that the shorthand is likely to be interpreted in multiple and sometimes problematic ways.

- Actively seek to engage with, understand and incorporate into soil-related research, the less traditional dimensions of capital, codification and connectivity. Avoiding discussions of soil as a property, and leaving the development of policy and the communication of information to others, is a likely route to losing control of the Soil Security discourse. Loss of control of the discourse may, possibly, lead to a superficial rhetorical shift from the old term of soil conservation to Soil Security, an embracing of a buzzword with no real changes in the practice or impact of soil science. Or, soil science may be securitized, with various consequences, including that the threats to specific human needs may override broader systemic imperatives.

- Seek and encourage forms of genuine collaboration. The second and third Soil Security conferences, and this Special Issue, take steps in this direction by inviting papers from non-soil science discipline areas, but there is much more that can be enabled and achieved by encouraging collaborative reflection and co-creation of futures.

- Commit resources, including time, to shared exploration of words, discourses and framings of their soil topics, rather than solely focusing on biophysical topics. The act of interrogating and understanding the discourse can be used to build collaborations, share understandings and design new futures.

- Reflect, individually and collaboratively, on the practice and outcomes of securing soil, to enable it to be a transformative, rather than constraining, frame.

## 6. Conclusions

Framing soil as Security issues risks treading an ever-narrowing path into control through rules, legislation and even force. Simultaneously, the Soil Security framing can facilitate opportunities to develop new and emerging understandings of the relationships between humans and soil. Which future comes about will be strongly influenced by the actions of soil science communities around the world. Engaging with the full scope of the metaphor is, therefore, necessary if Soil Security is to be an integrating, rather than defensive, operational frame.

**Funding:** This research received no external funding.

**Acknowledgments:** The work has been supported by the *Cooperative Research Centre for High Performance Soils* whose activities are funded by the Australian Government's Cooperative Research Centre Program.

**Conflicts of Interest:** The author declares no conflict of interest.

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
