# Peer review of "The Opportunities and Risks of the Soil Security Metaphor: A Review"

_sustainability, doi:10.3390/su11164464_

Round 1
Reviewer 1 Report
I cannot agree with the authors and still consider that the manuscript is more philosophical than biotechnological. Sorry I cannot say anything different.
Author Response
Response to reviewer 1 comments
Comment 1.
I cannot agree with the authors and still consider that the manuscript is more philosophical than biotechnological. Sorry I cannot say anything different.
Response
I agree with Reviewer 1 that this manuscript is philosophical rather than biotechnological. I do not think this means it should not be considered for publication in this journal.
Philosophy examines the fundamentals of knowledge and practice. I argue that biotechnological research needs to be understood within its the ontological framework(s) to be truly valuable. This view is reflected in the stated scope of Sustainability (my emphases added as bold text below)
Sustainability (ISSN 2071-1050) is an international and cross-disciplinary scholarly, open access journal of environmental, cultural, economic and social sustainability of human beings, which provides an advanced forum for studies related to sustainability and sustainable development. It publishes reviews, regular research papers, communications and short notes, and there is no restriction on the length of the papers.
Our aim is to encourage scientists to publish their experimental and theoretical research relating to natural sciences, social sciences and humanities in as much detail as possible in order to promote scientific predictions and impact assessments of global change and development. Full experimental and methodical details must be provided so that the results can be reproduced.
Reviewer 2 Report
By deconstructing the words ‘soil’ and ‘security’ the author uses language and metaphor to examine the meaning (both intended and unintended) conveyed by the term ‘Soil Security’.
General Comments:
While I found the manuscript on the whole interesting, there are aspects that I would suggest could be improved upon or delved into with more detail. Chief among these, the author states in the abstract “To guide this engagement, the review paper concludes with suggestions on how to reflect on the practice of soil science and its role in the future of humanity.” I found the conclusion somewhat lacking in this regard and could be expanded upon and specific suggestions detailed more fully.
Much of the manuscript discussion centred around the term ‘security’ in Soil Security, and I found that discussion interesting and enlightening. I would, however, suggest more exploration around the meaning and etymology of the term ‘soil’. In contrast to security I found the discussion around soil to be somewhat oversimplified (granted this is coming from someone with a “soil” science background). While I understand the point the author is making, I’m not sure I would agree with soil being considered an academic term. My suspicion is that the terminology may depend more on its use. For instance, I suspect most farmers or persons directly utilising the resource would use the term ‘soil’ whereas for instance a construction worker, who is primarily concerned with moving or shifting the earth out of the way, may simply see it as ‘dirt’. While the author notes that dirt has negative connotations, the term soil can have a negative connotation as well (e.g. to soil one’s clothes). Whether this particular train of thought is worth pursuing I leave up to the author, but I do think there is need for further discussion around the term soil.
A few specific comments:
Lns 35-36 a brief description of the 5 dimensions of soil security would be helpful, particularly as it feeds into figure 4.
Lns 188-189 Actually there is a degree of crossover between soil and water security. Soil degradation (such as erosion) or nutrient loss both have on site and off site consequences which potentially can cross jurisdictional boundaries and for instance effect water quality itself. Figure 4. Some
explanation needed around definition of the dimensions of soil security and how
they relate to owning/being safe/being in control - while some of these connections
are rather obvious, others are not. Figure 4. Some
explanation needed around definition of the dimensions of soil security and how
they relate to owning/being safe/being in control - while some of these connections
are rather obvious, others are not.
Reviewer 3 Report
The paper is very clear, well written and addresses a very important and timely topic, that of specific framings of discourses and policies regarding soil and soil conservation. Starting from the proposal of the concept of ‘soil security’ by a group of international leading soil scientists in the beginning of the 2010s, the author asks what consequences the framing of soil issues in terms of ‘soil security’ could have now and in the future. The paper relies on a very interesting approach to the use of metaphors and discourses in the environmental field, based on a significant literature. It argues that the framing in terms of ‘soil security’ may contribute to narrowing the approach to soil conservation in terms of national rules, defense, and force - which is contrary to the initial intentions of its proponents.
It is a convincing thesis, yet the argument could benefit from some more details and examples regarding how the term ‘soil security’ was further used by scientists and stake-holders (if it was indeed).
The author suggests at some point that the worst scenario would be for soil scientists not to use and elaborate on ‘soil security’ anymore (that would open the way for further uses by other stakeholders with different objectives in mind). Is this risk actual and documented, or is it a potential risk (yet no less relevant and real) related to the security metaphor?
Was the term only used by
soil scientists from the initial group that coined it? Or was it also
used by other actors and which ones, when and where? If it was not used
and expanded on by other people, maybe this would require some comments.
Also, some more details on the context and goals of the various texts (article, journal, book…) in which the term was proposed and elaborated by Koch, Mac Bratney et al. would also be welcome. Some more details on the social and political context in which the term was coined and proposed would allow for backing up the semantic analysis. In which context did the group of soil scientists choose this expression (and maybe if this information is available, what other options were at hand?)
I have 2 other minor comments.
In relation to the analysis of the environmental security discourses, it may be relevant to refer to Claudia Aradau’s writings on the expansion and signification of securitization framings for instance: Aradau, Claudia (2010), ‘Security that matters: critical infrastructure and objects of protection’, Security Dialogue vol. 41, no. 5:491-514.
In line with my first comment on potential further uses of the ‘soil security’ term and metaphor, I was wondering whether introducing some brief reflections on non-English languages, especially French, could be helpful. While ‘soil security’ seems to translate very directly in French into ‘sécurité du sol’, the French word ‘securité’ means the actual safety of people and bodies, and refers only very remotely to the notion of ‘being in control regarding the future’. I feel that the French translation is still closer to the notion of State-related rules, defense or even army force, than the English term (by the way I am not aware of many occurrences of ‘sécurité environnementale’ in French environmental policies or governance).
A couple sentences need corrections:
p.2 line 65: “Metaphors are usually formulated such that a target idea, is/as a source” needs clarification
p.4 line 65: “securitized language” : or ‘securitization language’?
p.4 line 166: “Related, formulations”: remove the comma
p. 5 line 210: “metaphors have been shown provide”: ‘to’ is lacking
Round 2
Reviewer 2 Report
The author has adequately addressed this reviewer's comments. I would recommend the article be accepted for publication.